# Drepmel—A Multi-Omics Melanoma Drug Repurposing Resource for Prioritizing Drug Combinations and Understanding Tumor Microenvironment

**DOI:** 10.3390/cells11182894

**Published:** 2022-09-16

**Authors:** Zachary J. Thompson, Jamie K. Teer, Jiannong Li, Zhihua Chen, Eric A. Welsh, Yonghong Zhang, Noura Ayoubi, Zeynep Eroglu, Aik Choon Tan, Keiran S. M. Smalley, Yian Ann Chen

**Affiliations:** 1Biostatistics and Bioinformatics Shared Resource, H. Lee Moffitt Cancer Center and Research Institute, Tampa, FL 33612, USA; 2Department of Biostatistics and Bioinformatics, H. Lee Moffitt Cancer Center and Research Institute, Tampa, FL 33612, USA; 3Department of Dermatology and Cutaneous Surgery, University of South Florida, Tampa, FL 33612, USA; 4Department of Cutaneous Oncology, H. Lee Moffitt Cancer Center and Research Institute, Tampa, FL 33612, USA; 5Department of Tumor Biology, H. Lee Moffitt Cancer Center and Research Institute, Tampa, FL 33612, USA

**Keywords:** multi-omics, melanoma, drug repurposing, microenvironment

## Abstract

Although substantial progress has been made in treating patients with advanced melanoma with targeted and immuno-therapies, de novo and acquired resistance is commonplace. After treatment failure, therapeutic options are very limited and novel strategies are urgently needed. Combination therapies are often more effective than single agents and are now widely used in clinical practice. Thus, there is a strong need for a comprehensive computational resource to define rational combination therapies. We developed a Shiny app, DRepMel to provide rational combination treatment predictions for melanoma patients from seventy-three thousand combinations based on a multi-omics drug repurposing computational approach using whole exome sequencing and RNA-seq data in bulk samples from two independent patient cohorts. DRepMel provides robust predictions as a resource and also identifies potential treatment effects on the tumor microenvironment (TME) using single-cell RNA-seq data from melanoma patients. Availability: DRepMel is accessible online.

## 1. Introduction

Cutaneous melanoma is the deadliest form of skin cancer, with a tendency to aggressively metastasize to multiple organs [1]. Melanoma has long been a poster child for personalized medicine with targeted therapies such as the BRAF inhibitors and the BRAF-MEK inhibitor combination being highly effective against the 50% of melanomas with activating BRAF mutations. Despite this, targeted therapies are lacking against NRAS mutant melanoma and the approximately 25% of melanomas that have no identified oncogenic driver mutations. Recently, immunotherapies such as anti-PD1 and anti-CTLA4 have shown great promise to improve patient outcomes. However, after treatment failure, limited treatment options are available. Drug combinations have been developed and approved for overcoming treatment resistant to targeted and immunotherapies, yet computational methods and resources have been limited for predicting drug combinations. Here, we developed three multi-omics approaches to predicting effective combination therapies using two independent cohorts of melanoma patient data, and the results are displayed through a user-friendly Shiny app DRepMel. Machine and deep learning approaches have been shown to be powerful for predicting anti-cancer combination therapies using cancer cell line data [2,3,4]. To assess potential treatment effects of targeted and immunotherapies, we used omics data from melanoma patients which consist of both tumor and immune/stromal cells instead of cancer cell line datasets. Furthermore, targeted tumor immune microenvironment (TME) derived from scRNA-seq data of melanoma patient samples [5] was included in the app for understanding the potential combination therapy impact on TME.

## 2. Methods

We developed an integrative approach for drug repurposing and predicting combination therapies for melanoma patients and applied them to two independent melanoma patient cohorts with matching Whole Exome Sequence (WES) and RNA-seq datasets from the same patients: the TCGA (N = 459) and Moffitt Melanoma cohorts (N = 135; Figure 1). The Moffitt Melanoma Cohort (N = 135) was described in our previous work [6]. Briefly, this study (MCC# 19147) was conducted in accordance with recognized ethical guidelines (e.g., Declaration of Helsinki, CIOMS, Belmont Report, U.S. Common Rule) and was approved by Chesapeake Institutional Review Board (IRB). A waiver of consent was granted by Chesapeake IRB. We include additional information on Whole Exome Sequence Analyses (WES) and RNA-seq analyses here. Summary information is included in Table 1.

### 2.1. WES and RNA-Seq Sequence Analyses

WES data has been generated for tumors and matched normal samples, with a depth of coverage averaging around 100×. Sequence reads were aligned to the reference human genome (hs37d5) with the Burrows-Wheeler Aligner (BWA) [7], and insertion/deletion realignment and quality score recalibration were performed with the Genome Analysis ToolKit (GATK) [8]. Tumor-specific mutations were identified with Strelka [9] and MuTect [10], and were annotated to determine genic context (i.e., non-synonymous, missense, splicing) using ANNOVAR [11]. Additional contextual information was incorporated, including allele frequency in other studies such as 1000 Genomes and the NHLBI Exome Sequence Project, in silico function impact predictions, and observed impacts from databases like ClinVar (http://www.ncbi.nlm.nih.gov/clinvar/) (accessed on 1 March 2016), the Collection Of Somatic Mutations In Cancer (COSMIC), and The Cancer Genome Atlas (TCGA). Mutation signatures (alterations across possible trinucleotide sequences) were counted and derived as described in Alexandrow et al. [12] as implemented by deconstructSigs [13]. WES quality control includes read metrics following each analysis step (fraction duplicate reads, fraction mapped reads), depth of coverage assessment across the targeted regions, and common genotype comparisons across samples to ensure proper sample matching. Mutations were counted as follows: observed in Strelka-specific OR (MuTect AND Strelka-sensitive), predicted to be protein altering, and <1% frequency in 1000 Genomes.

RNA-seq data has also been generated on the same tumor samples. Sequence reads were aligned to the human reference genome in a splice-aware fashion using Tophat2 [14], allowing for accurate alignments of sequences across introns. Aligned sequences were assigned to exons using the HTseq package [15] to generate initial counts by region. Normalization, expression modeling, and difference testing were performed using DESeq [16]. RNAseq quality control includes in-house scripts and RSeqC [17] to examine read count metrics, alignment fraction, chromosomal alignment counts, expression distribution measures, and principal components analysis and hierarchical clustering to ensure sample data represents experiment design grouping. TCGA whole exome data was downloaded from the NIH TCGA website in March 2016 [18]. The MAF file was converted to VCF, and then annotated as described for the TCC data. Mutations were counted as follows: predicted to be protein-altering and <1% frequency in 1000 Genomes. Level 3 of RNA-seq data was used in this study. RNA-seq expression data was de-batched between the TCGA and Moffitt cohorts using the Combat function in the sva package in R [19].

### 2.2. Doublet Combination Therapy Candidates

For treatment predictions, a total of 5894 treatments and their target genes from Drug SIGnatures DataBase DSigDB, [20], selleckchem.com (accessed on 1 March 2016), and commonly known immune checkpoint therapies were included [20] as therapy candidates. Among these, 5845 drugs were from DsigDB, 38 HDAC inhibitors were from selleckchem.com, and 11 drugs were known immune checkpoint therapies.

For initial screening, single therapy analyses were performed for each candidate therapy to identify plausible “seed” therapies to form the pool of doublet combination candidates. A single therapy analysis consists of three parts and was summarized using Fisher’s Product method (FPM). To assess the potential efficacy of every single therapy, the analyses via mutation, expression, and patients’ overall survival (OS) were used as a surrogate for clinical outcome. RNAseq and mutation status of target genes of a therapy were the primary independent variables in respective Cox PH models, adjusting for age and *IPI/NIVO* and *BRAF* treatment. The single therapy models to evaluate the association between mutation and OS is defined as $S_{MUT}\left(t\right)=\beta_{M}x+\beta_{a}age+\beta_{B}BRAF+\beta_{I}IPI/NIVO$, where x is an indicator of mutations in the target genes of the candidate drug. *BRAF* is an indicator of *BRAF* inhibitor treatment and *IPI/NIVO* is an indicator of any check point inhibitors. To evaluate the association between the gene expression data and survival the model is $\;S_{PC}\left(t\right)=\beta_{E}x_{PC}+\beta_{a}age+\beta_{B}BRAF+\beta_{I}IPI/NIVO$, where $x_{pC}$ is the first principal component (PC1) of the gene expression data of the target genes of the candidate drug. Since a drug often targets multiple genes, a principal component analysis was used to summarize and reduce the dimensionality of the gene expression data for genes targeted by a drug. PC1 explains the maximal amount of variance of the expression data from a drug set and was used in the survival and eQTL analyses. The eQTL analysis was performed using the Wilcoxon rank sum test using the PC1 of the expression values of the target genes and the mutation indicators in the target genes. When there is a mutation detected in at least one of the target genes of a drug (set) for a patient, the mutation status for this patient is coded as 1 (in the binary 0/1 coding) for the eQTL analysis. When no mutation is found in the target genes from a drug (set), then it is coded as zero. PC1 was used in the Wilcoxon rank sum test for evaluating the potential association between the mutations and expression of the target genes within a drug set. FPM was first used to synthesize the results from three analyses for each candidate drug within each cohort. Then, FPM was used to generate a summary of the results from the two cohorts. *p*-values associated with the Chi-squared statistic from the FPM were used to prioritize the single treatment. A false discovery rate (FDR) was used to adjust for multiple comparisons. The treatments with FDR < 0.05 were selected as “seed” therapies to pair with each of the remaining treatments to form the doublet candidate pool. There are 37 therapies with FDR < 0.05: 26 FDA-approved Kinases, 7 Immuno drugs, and 4 HDAC drugs. We also included two clinically used treatments for melanoma patients: Panobinostat and Trametinib as part of 39 seed treatments to formulate the doublet pool. Pairing each of the seed treatments with each of the remaining treatments from the 5848 candidates results in a total of 73,007 combinations in the doublet pool.

### 2.3. Drug Repurposing Models for Doublets

To assess the potential treatment effects of each doublet therapy candidate, the association between mutation, expression, and patients’ overall survival, used as a surrogate for clinical outcome, was examined. RNAseq and mutation status of target genes within a therapy were the primary independent variables in respective Cox PH models, adjusting for age and *IPI/NIVO* treatment and *BRAF* treatment (Equations (1) and (2)). The treatment interaction was also included in the model. An expression quantitative trait loci (eQTL) analysis was performed to assess the potential transcriptional impact of the mutations using a Wilcoxon test (Equation (3)). Since actual mechanisms of treatment action through DNA, RNA or their interaction is uncertain, three models/methods were formulated based on different assumptions. Method 1 (Equation (4)) combines all evidence from (Equations (1)–(3)). Method 2 evaluates the evidence from gene expression (Equation (5)) while Method 3 evaluates the most significant evidence with a minimum *p*-value among the 3 sets of evidence (Equations (1)–(3)) within each cohort (Equation (6)). Details are described below. Fisher’s Product method was used to combine evidence from two cohorts. Further filtering (*p* < 0.05) for each cohort was performed for Method 2 and summarized in the Shiny App DRepMel. Subset analyses were performed for patients with BRAF, NRAS mutations, or Triple WT cohorts.

To evaluate the association between patients’ overall survival and somatic mutation (MUT), gene expression using the PC1, and the potential mutation impact on target genes, the following three equations were formulated. To evaluate patients’ survival with a mutation in target genes of each doublet:(1)$$S_{MUT}\left(t\right)=\beta_{M1}x_{1}+\beta_{M2}x_{2}+\beta_{M3}x_{1}x_{2}+\beta_{M4}age+\beta_{M5}BRAF+\beta_{M6}IPI/NIVO$$
where $x_{1}$ and $x_{2}$ are indicators of mutations in the target genes of drugs 1 and 2, respectively. *BRAF* is an indicator of BRAF inhibitor treatment and *IPI/NIVO* is an indicator of any checkpoint inhibitors. To evaluate patients’ survival with expression in target genes of each doublet:(2)$$S_{PC}\left(t\right)=\beta_{E1}x_{1}+\beta E_{E2}x_{2}+\beta_{E3}x_{1}x_{2}+\beta_{M4}age+\beta_{M5}BRAF+\beta_{M6}IPI/NIVO$$
where $x_{1}$ and $x_{2}$ are the first principal component of the gene expression data of the target genes of drugs 1 and 2, respectively. To assess the potential functional impact of mutations on gene expression, the *eQTL* is performed with the Wilcoxon rank sum test. The PC1 of the expression values of the target genes was used with the mutation indicators in the target genes of both drugs.
(3)$$eQTL:PC1_{1}\sim \;x1\;of\;target\;genes\;in\;drug\;1\;PC1_{2}\sim \;x2\;of\;target\;genes\;in\;drug$$

To enhance the robustness of the inference, analyses were performed using two independent melanoma patient cohorts: TCGA (N = 459) and Moffitt Melanoma cohorts (N = 135). Fisher’s Product method was used to synthesize the results from each of the analyses above (Equations (1)–(3)) with the following notation: p(β) is the *p*-value of the coefficient β in equations 1 or 2 or *p*-value of eQTL analysis (3).

M = “Mutation” model, E = Expression model.

“1” = drug 1, “2” = drug 2

“t” = TCGA cohort, “m” = Moffitt cohort

Since the actual mechanisms of treatment action through DNA, RNA or their interaction is uncertain, three models were formulated based on different assumptions.

Method 1 combines all evidence from Equations (1)–(3):

Meta P (It includes all eight terms from both cohorts)
(4)$$\;\chi_{2k\;}^{2}\sim -2\sum (\begin{array}{c} \ln\left(P\left(\beta_{M1t}\right)\right)+\ln\left(P\left(\beta_{M2t}\right)\right)+\ln\left(P\left(\beta_{M3t}\right)\right)+\ln\left(P\left(\beta_{E1t}\right)\right)+\ln\left(P\left(\beta_{E2t}\right)\right)+\ln\left(P\left(\beta_{E3t}\right)\right)+\ln\left(P\left({eQTL}_{1t}\right)\right)+\ln\left(P\left({eQTL}_{2t}\right)\right)\\ +\ln\left(P\left(\beta_{M1m}\right)\right)+\ln\left(P\left(\beta_{M2m}\right)\right)+\ln\left(P\left(\beta_{M3m}\right)\right)+\ln\left(P\left(\beta_{E1m}\right)\right)+\ln\left(P\left(\beta_{E2m}\right)\right)+\ln\left(P\left(\beta_{E3m}\right)\right)+\ln\left(P\left({eQTL}_{1m}\right)\right)+\ln\left(P\left({eQTL}_{2m}\right)\right) \end{array})$$

Method 2 evaluates the association evidence using expression data alone: CombinedPExpression (It includes the main effects from expression Equation (2)).
(5)$$\chi_{2k\;}^{2}\sim -2\sum \left(\ln\left(P\left(\beta_{E1t}\right)\right)+\ln\left(P\left(\beta_{E2t}\right)\right)+ln\left(P\left(\beta_{E1m}\right)\right)+ln\left(P\left(\beta_{E2m}\right)\right)\right)$$

Since the mechanism of action could differ among treatments so, within each cohort, the minimum *p*-value of the test among 3 tests (Equations (1)–(3)).

Method 3 combines the minimum *p*-value of the tests between two cohorts. For example, if Min *p*-value is from the same model, say the $S_{MUT}\left(t\right)$ models in TCGA and Moffitt cohorts, then the interaction term *p*-values are used.
(6)$$\chi_{2k}^{2}\sim -2\sum \left(\ln\left(P\left(\beta_{M3t}\right)\right)+\ln\left(P\left(\beta_{M3m}\right)\right)\right)$$

### 2.4. Potential TME Targeted by the Predicted Doublet Therapies

Potential TME targeting by the predicted doublet therapies was inferred using the scRNA-seq data of 28,078 single cells from 43 patient samples [5]. All analyses and the Shiny App were performed and implemented using R.

## 3. Web Application and Results

The predicted doublets by Method 2 of version 1.0 of the Drepmel are available for visualization using the Shiny application at http://drepmel.moffitt.org/, which will be maintained for at least 3 years (contact zachary.thompson@moffitt.org or ann.chen@moffitt.org with technical issues). The R code defining the server-side logic of the Shiny application is available in Appendix A. The R code controlling the layout and appearance of the application is available in Appendix A.

The input for the app includes two drop-down menus of treatments and radio buttons to choose the (sub-) group of patients corresponding to the major melanoma genotypes (All, NRAS, BRAF, Triple WT). Two additional drop-down menus provide target genes to select in each treatment for single gene expression heatmaps to understand the potential treatment effect in the TME.

The Shiny application includes a tab for introduction, method, and the following results tabs:Tables of top doublet combinations summarized the overall results and those for each of the major melanoma genotype groups.TME: Heatmap and Violin Plot Highlight Potential Targeted Cells Populations by Each TherapyThe mutation and survival tab displays Kaplan–Meier plots of overall survival based on mutation status in the target genes of the selected doublets in each cohort.The PC1 and survival tab shows the tables of genes and PC1 loadings in the target gene sets of each treatment for each cohort along with the KM plots of PC1 and overall survival for each treatment in both cohorts. The PC1 values are dichotomized at the median.The eQTL tab displays the box plots of gene expression in both cohorts by mutation status in the target genes. It also displays the summary statistics of the de-batched expression on a log scale.

The results from Methods 1, 2, and 3 are included in Appendix A. The top combinations include plausible candidates. For the overall analyses, the top combinations include known effective treatments (anti-PD1), plausible ones (Lag3, nilotinib), and additional treatment combinations which could be further investigated (Appendix A). Robust findings between the TCGA and Moffitt cohorts for patients with limited treatment options (NRAS or triple WT patients) provides a short list of candidates for further investigation. The 52 predicted combinations for the NRAS subgroup contain some interesting candidates. The top candidate combining LAG3 and clioquinol show consistent finding between two patient cohorts (Figure 2). This combination, while unexpected, could offer some novel avenues for melanoma therapy. Clioquinol has effects on the proteasome as well as copper and zinc metabolism, and has the potential to alter transcriptional activity in both cancer cells and immune cells [21,22]. It is possible that these broadly targeted effects on the tumor transcriptional state could increase sensitivity to more broadly used immunotherapies, such as the anti-LAG3 antibody. Our group has already demonstrated that immunotherapies can be used in sequence with targeted therapies to deliver long-term anti-tumor effects in mouse melanoma models. In this instance, these effects are driven both by modulation of signaling in the tumor and reprogramming of the immune microenvironment [23]. Rigorous pre-clinical evaluation of the drug combinations selected from tools such DRepMel could lead to a robust pipeline of repurposed drug combinations for future clinical evaluation. The TME results indicate that each therapy is likely targeting different cell populations: lymphoid and myeloid, respectively. This provides insight into how the candidate combinations might work. Robust predictions are also provided for the BRAF subgroup of melanoma patients (Figure 3). The tool DRepMel provides a useful computational resource with robust findings for hypothesis generation. It also yields insights on potential treatment impacts on the TME for further investigation.

## Figures and Tables

**Figure 1 cells-11-02894-f001:**
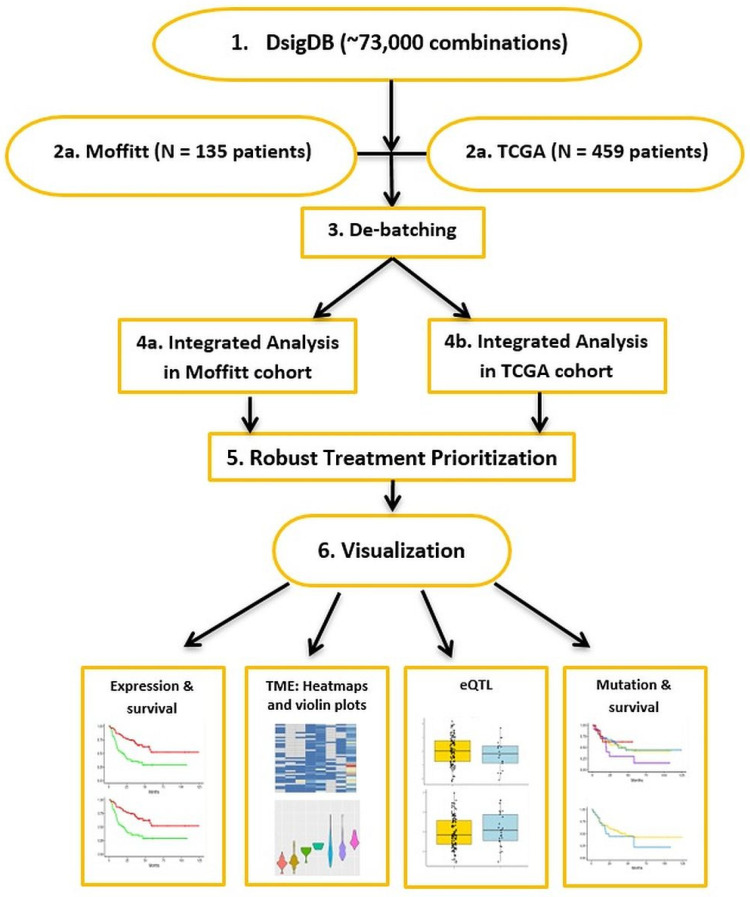
The workflow of DRepMel: an integrative multi-omics drug repurposing approach for predicting combination therapies for melanoma patients using two independent melanoma patient cohorts (with matching Whole Exome Sequence (WES) and RNA-seq datasets from the same patients: the TCGA (N = 459) and Moffitt Melanoma cohorts (N = 135)).

**Figure 2 cells-11-02894-f002:**
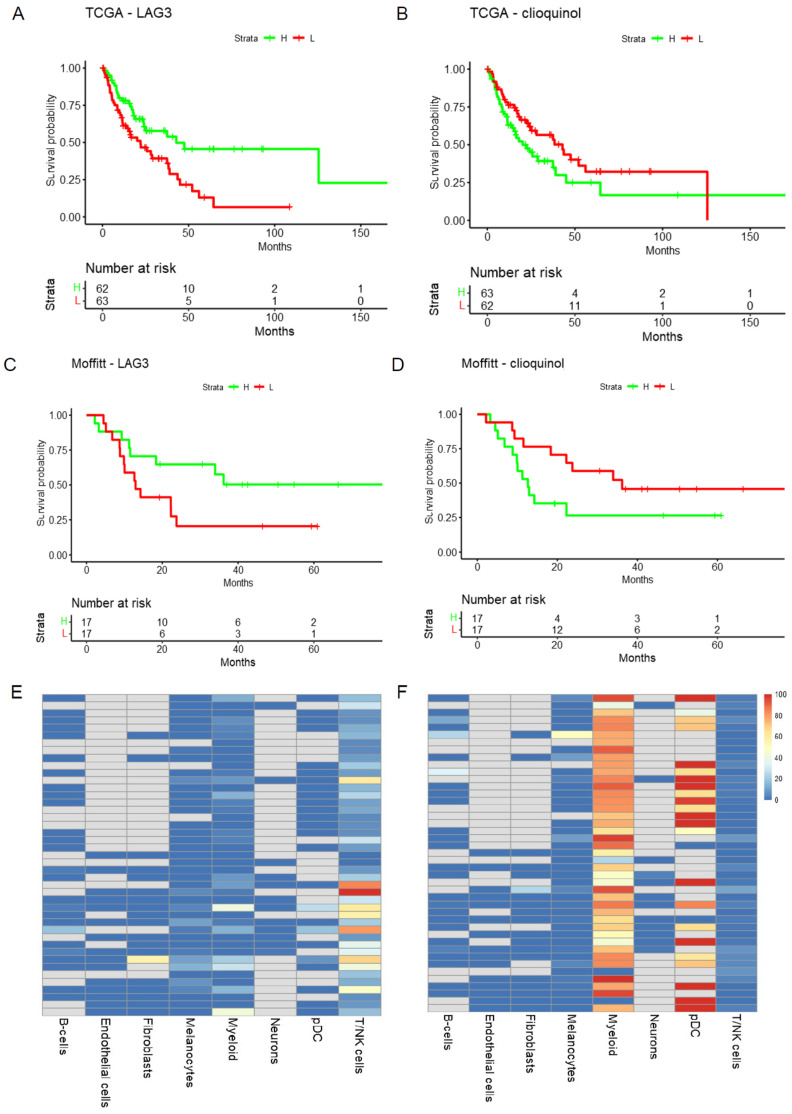
DRepMel predicts that the combination treatments of LAG3 and clioquinol (HL60 down) could be considered for melanoma patients with NRAS mutations. The signals are robust and consistent in the (**A**,**B**) TCGA and (**C**,**D**) Moffitt cohorts. (**E**,**F**) Each therapy likely targets different cell populations, i.e., T/NK and myeloid, as shown using scRNA-seq data.

**Figure 3 cells-11-02894-f003:**
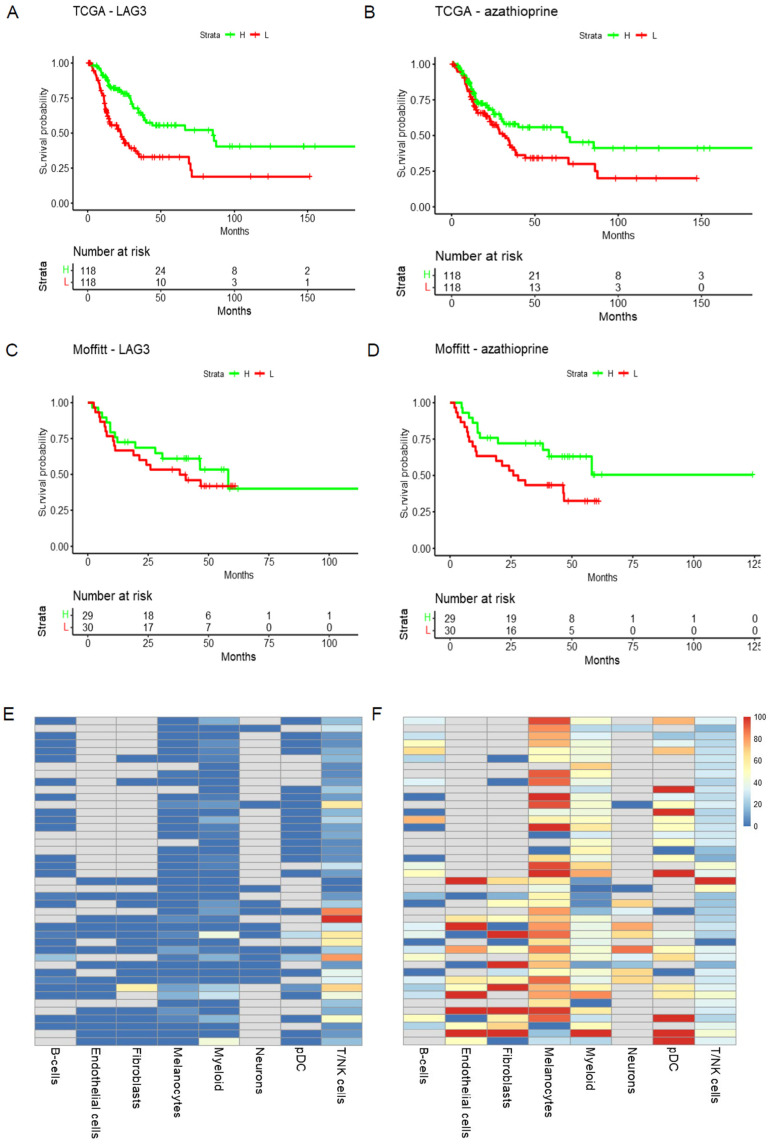
DRepMel predicts that the combination treatments of LAG3 and Azathioprine (MCF7 down) could be considered for melanoma patients with BRAF mutations. The signals are robust and consistent in the (**A**,**B**) TCGA and (**C**,**D**) Moffitt cohorts. (**E**,**F**) Each therapy likely targets different TME compartments.

**Table 1 cells-11-02894-t001:** Information on sample size, available meta-data, and DNA/RNA sequencing of the TCGA and Moffitt cohorts.

		TCGA (N = 459)	Moffitt (N = 135)
Mutation Cohorts n (%)			
	BRAF	236 (51.4)	59 (43.7)
	NRAS	125 (27.2)	34 (25.2)
	Triple Wild type	88 (19.2)	28 (20.7)
Age mean (sd)		61.5 (15.0)	62.7 (15.3)
IPI/NIVO treatment n (%)		15 (3.6)	51 (38.8)
BRAF treatment (%)		5 (1.1)	29 (21.5)
Gender n (%)			
	Female	175 (38.1)	49 (36.3)
	Male	284 (61.9)	86 (63.7)
Transcriptomics		RNA-seq	RNA-seq
DNA mutation		WES	WES

## Data Availability

Data and results generated during this study are available as Appendix A and available by request through the website. All R codes are uploaded as Appendix A, server.R and ui.R. A user manual is also available through the website.

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
