# Peer review of "Drepmel—A Multi-Omics Melanoma Drug Repurposing Resource for Prioritizing Drug Combinations and Understanding Tumor Microenvironment"

_cells, 2022, doi:10.3390/cells11182894_

Round 1

Reviewer 1 Report

  1. Please include a table showing the sample size of each cohort and other meta-data available, e.g. technology used to generate data (exome or scRNAseq), mutation status, median age etc. 
  2. Line 92/- please include the file hyperlink as a citation under the references section. A reference manager will help.
  3. Please cite the combat package.
  4. It's not clear how eQTLs were identified. Please include relevant details on how genetic variants and gene expression were associated with eQTL prediction.
  5. Line 103, please describe the criteria for “initial screening”. 
  6. Please redraft the method section. It is difficult to understand the flow of methods, e.g. equations are referred to first and then equations are described in a subsequent section.  I would suggest to three sub-sections, one for each method (i.e. method1/2/3). Instead of the method, you might want to use a different term, e.g. Pipeline1/2/3.                                
  7. It is not clear how PCs were calculated and why PCs were used. What information PCs are adding?
  8. The entire section- “Doublet Combination Therapy Candidates”, needs to be presented in a clear manner. A lot of things are going on but they all look unconnected while reading.                         
  9. The results section needs more details. The results are interesting and provide novel insight into combinatorial drug therapy but details and discussion are missing.     

Reviewer 2 Report

The manuscript by “Thompson et.al.” provide the rationale for to use of combination therapy to treat drug resistance melanoma. In the manuscript they use the Shiny app, DRepMel to provide rational combination treatment predictions for melanoma patients. The overall manuscript is well written and provides evidence to use this software to predict combination treatments for melanoma.  

Author Response

We thank reviewer 2 for spending the time to review the manuscript and for their positive feedback.

Reviewer 3 Report

Necessary information is given in the introduction section about the subject of the article and the method part is explained in detail. Necessary explanations have been made in the Web Application and Results section. Important information is given for future studies on the subject. I think it would be appropriate to publish the article.

Author Response

We appreciate reviewer 3’s time and effort to review the manuscript and are encouraged they found our work appropriate for publication.